# Demonstrating the processes and outcomes of a rural Community Mental Health Rehabilitation Service: A realist evaluation

A. Leet[1☋‡], S. Dennis[2☋‡], J. Muller[3☋‡], S. Walsh[3☋‡], H. Bowen-Salter[4☋], J. Kernot[5☋‡]*

1 Barossa Hills Fleurieu Local Health Network/ Rural and Remote Mental Health Service, Angaston, Australia, 2 Flinders and Upper North Local Health Network/ Whyalla Integrated Mental Health Service, Whyalla, Australia, 3 Department of Rural Health, Allied Health and Human Performance/ University of South Australia, Adelaide, Australia, 4 International Centre for Allied Health Evidence, Allied Health and Human Performance/University of South Australia, Adelaide, Australia, 5 Occupational Therapy Program, Allied Health and Human Performance/ University of South Australia, Adelaide, Australia

☋ These authors contributed equally to this work.
‡ AL, SD, JM, SW & JK also contributed equally to this work.
* Jocelyn.Kernot@unisa.edu.au

**Data Availability Statement:** All relevant data are within the manuscript and/or Supporting Information files.

## Abstract

### Background

As part of significant mental health reform, the Community Mental Health Rehabilitation Service (CMHRS) was implemented in rural South Australia. The CMHRS is a 10-bed mental health residential program offering rehabilitative mental health support to rural residents.

### Aim

To analyse the CMHRS service delivery model and its impact on recovery outcomes for consumers.

### Methods

A mixed method, realist evaluation approach was utilised. A purposive sample of CMHRS staff (n = 6) and consumers (n = 8) were recruited. Consumer recovery was measured using the RAS-DS (on admission and discharge). Participants' perspectives of the service were gained via one staff focus group (n = 6) and individual semi-structured interviews (consumers n = 6; staff n = 2). Pre-post RAS-DS scores were analysed using paired t-tests/Wilcoxon paired-signed rank test, with qualitative data analysed thematically.

### Results

Significant positive increases in RAS-DS total scores were observed at discharge, supported by the qualitative themes of (re)building relationships and social connections and recovering health and wellbeing. Contextual factors (e.g. staffing) and program mechanisms (e.g. scheduling) impacting on service implementation were identified.

**Funding:** This project was supported by a University of South Australia and South Australian Health, Allied Health Research Collaboration Grant. The funders had no role in study design, data collection and analysis, decision to publish, or preparation of the manuscript.

**Competing interests:** The authors have declared that no competing interests exist.

## Conclusion

Maintaining a rehabilitation recovery-focused approach, balanced with an appropriately trained multi-disciplinary team, are vital for maximising positive consumer outcomes.

## Significance

This realist evaluation identifies critical factors impacting rural mental health rehabilitation service delivery.

## Introduction

Mental and substance use disorders are the third largest cause of total disease burden [1], with people who experience mental illness more likely to have adverse health and social outcomes [2]. One in five Australians experience mental ill-health each year, with an estimated health expenditure of 98.8 billion dollars per annum [3]. Rural Australians are exposed to higher risk factors for mental health problems including unemployment, and physical health issues [4, 5], with hospital admission rates for mental health conditions, intentional self-harm, suicide, and drug and alcohol problems increasing with remoteness [4]. However, despite increased risk factors, access to specialist mental health services is lower compared to major cities [4]. To address this inequity, there is a need for improved access to mental health services inclusive of rehabilitative mental health support in rural settings, including evaluation and improvement of existing mental health services [6].

This paper outlines a realist evaluation of a rural Community Mental Health Rehabilitation Service (CMHRS), a clustered housing (1–3 people per house) residential program offering rehabilitative mental health support to rural South Australians. The maximum number of consumers that can be in the program at one time is 10. A realist evaluation approach views programs as theories as "they are 'embedded', they are 'active', and they are parts of 'open systems'" [7, 8]. Given the need for evidence based mental health services in rural settings, this approach facilitates a detailed examination of an existing service including its contexts, mechanisms, and outcomes (CMO). This can help to inform future program design/improvements, implementation, and evaluation [8]. To elaborate further—'context' are the conditions in which the service operates; 'mechanisms' are the components or steps that lead to change; and 'outcomes' are the intended and unintended impact or consequences of the service (these are multifaceted and should include a range of output and outcome measures) [7, 8]. The specific aim of this study was to analyse the current service delivery model of the CMHRS and its impact on consumers recovery by answering the following questions:

- What do consumers and staff consider are the critical aspects of implementation, staffing, and organisational structure which influence how the CMHRS operates?

- How does the service impact on consumer recovery?

## Methods

### CMHRS

The South Australian (SA) Health (2012) Framework for Recovery Orientated Rehabilitation in Mental Health Care, notes that rehabilitation is about developing new skills

alongside relearning old skills, across all domains of a person's life through a stepped model [9]. The CMHRS is part of the SA Health stepped model of care, which provides graduated/tiered levels of care including secure care, acute care, intermediate/sub-acute care, rehabilitation, and supported accommodation. The CMHRS provides supported accommodation which aims to assist consumers to achieve and enhance independent living skills [9]. Referrals for the CMHRS can be received from all adult public mental health services across South Australia. Consumers eligible for CMHRS, generally have a serious mental illness and identified rehabilitative needs and goals (see S1 File which provides further details about eligibility criteria and consumer profile). A multidisciplinary team engage collaboratively with residents to develop an Individual Rehabilitation Plan (IRP), with a focus on skills and strategies to enable independence, community participation, and improved wellbeing. IRPs are dynamic and formally reviewed 6-weekly. The Recovery Assessment Scale-Domains and Stages (RAS-DS) is undertaken as part of the initial and ongoing assessment process [10]. The time that a resident remains engaged in the program depends on individual circumstances. Generally, the length of stay for a resident is anticipated to be up to 12 months, with the typical stay between 3–9 months. Transition from the service is discussed early in the residents' stay and is focused on the individual's goals. Activities provided as part of the program include weekly groups (psychoeducation, functional skill development, and social and recreational activities), one on one sessions with staff, and structured independent time.

## Study design

A realist approach was chosen for this study as it allows investigation of complex service delivery models and the development of a 'Middle Range Theory' (MRT), which can identify what program attributes are needed to promote effectiveness [7]. To assist this process, a preliminary program theory is developed using a Context-Mechanism-Outcome (CMO) configuration that can be considered during evaluation. In this case, a preliminary theory was developed from documentation of the service (Fig 1). Mixed method designs are recommended in a realist evaluation approach allowing the processes and impacts of the program to be evaluated [7]. This study included focus groups and semi-structured interviews with CMHRS staff, semi-structured interviews with consumers, pre-post RAS-DS scores, and case note audits. The findings of the evaluation were intended to identify areas for service improvement and refinement.

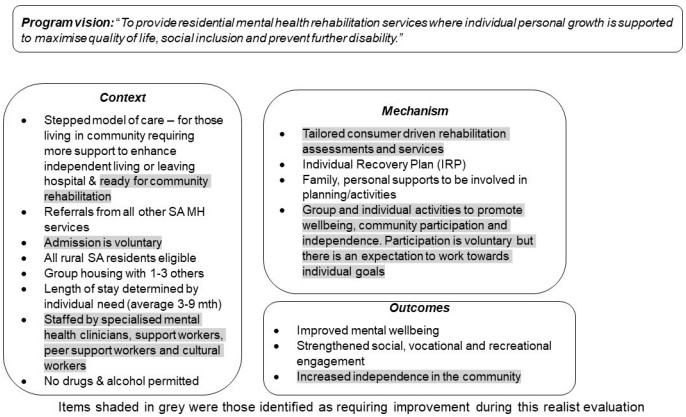

**Fig 1. Preliminary program theory.**

## Procedures

Ethical approval for this study was obtained by the SA Health Human Research Ethics Committee (protocol no HREC/18/SAH/118) and the University of South Australia's Human Research Ethics Committee (protocol no 202414). Formal written consent was gained from all participants prior to their involvement in the study.

**Staff focus groups/interviews.** A purposive sample of CMHRS staff from diverse professional backgrounds, were recruited via email invitation (which was sent to all staff members). To be eligible to participate, staff had to be English speaking, over 18 years of age, and currently working in the service. Staff participated in a focus group with optional individual interviews. The focus group was an hour in duration, with individual interviews ranging from 30–45 minutes. The purpose was to gather staff perceptions of: CMHRS individual and group-based intervention/strategies; intensity of service delivery; staffing and staff roles; and barriers and enablers to service delivery. A semi-structured interview guide was used to facilitate the focus group and interviews. The interview guide (S2 File) was piloted with two health staff who were not involved with the CMHRS. Minor amendments were made based on the feedback received. These were primarily regarding wording and flow of the questions.

The focus group and interviews were recorded and transcribed verbatim. NVivo software (Version 12; QSR International) was used to explore the qualitative data and to undertake thematic analysis. One member of the evaluation team (JM) conducted the focus group and interviews. The process of thematic analysis was guided by Braun and Clarke's [11] six phases of thematic analysis: 1) familiarisation with the data through detailed reading of the transcripts (JM & JK); 2) generating codes using NVivo software (JM & JK); 3) searching for themes (JM & JK); 4) reviewing and 5) defining themes through meetings and discussion with the research team (all authors) and 6) producing a report (all authors). Differences in opinion at all stages were openly discussed (between the team members involved) during face-to-face meetings and resolved through consensus.

**Consumer semi-structured interviews.** To be eligible to participate in the study, consumers had to be aged over 18 years, able to give informed consent, English speaking, and be a current service user or have used the service within the last six months. Current service users were recruited via short verbal presentation at the weekly community meeting and were provided with a participant information sheet and a consent form and given the opportunity to ask questions. A research team member attended the meeting the following week to obtain written consent. Previous service users discharged in the past six months were invited to participate in the study by their rehabilitation (case) coordinator.

Semi-structured qualitative interviews (SI 2) were conducted at, or following, discharge to gain consumers perspectives of the CMHRS. Interviews occurred at the CMHRS accommodation, local health service, or via telephone. Interviews were designed to gather information regarding participants perception of the intervention they had received during their time with the service, including one-on-one and group sessions, use of free and unstructured time, and how these impacted on their recovery.

The consumer and staff data were analysed separately initially (Braun & Clarke phases 1–3) and then finding brought together when the authors were reviewing and defining the themes (Braun & Clarke phases 4–6). Findings were categorised and reported using the realist evaluation Context-Mechanism-Outcome configuration.

**RAS-DS.** The Recovery Assessment Scale–Domains and Stages (RAS-DS) was used to evaluate recovery outcomes [10]. The CMHRS service implements the RAS-DS as part of their standard practice on admission and discharge. The RAS-DS is a 38-item self-administered questionnaire allowing consumers to rate their recovery in four domains Doing Things I

Value (6 Items), Looking Forward (18 Items), Mastering My Illness (7 Items) and Connecting and Belonging (7 Items). It uses a 4-point Likert scale with scores for each item added to gain a total score out of 152 (the higher the score, the higher the level of perceived recovery). The RAS-DS has demonstrated strong internal and construct validity and reliability (r = .42 to .70; Cronbach's α = .93) and has been well received by consumers and clinicians alike [12]. A recent study indicated that the RAS-DS is sensitive to detect change over time [13]. Pre-post (on admission and discharge) scores for the RAS-DS (total raw scores and raw scores for each of the 4 domains) were analysed using paired t-tests (normally distributed data) and Wilcoxon pair signed rank tests (for data that was not normally distributed). Normality of data were determined using Shapiro-Wilk test of normality and histograms. Data for total raw scores and scores on 3 domains were normally distributed. Data on the 'Doing Things I Value' domain was not normally distributed (pre-test scores .003 on Shapiro-Wilk test).

**Case note audit.** To describe consumers interaction with the service, an audit was undertaken of participants' CMHRS case notes. Case notes were examined to determine duration of stay, number and types of services received, and support required (when accessing these services). Pre-post support required (on admission and discharge) were analysed using Wilcoxon pair signed rank tests as data was not normally distributed (with a post-test score of .015 on Shapiro-Wilk test)

## Results

Utilising a concurrent triangulation mixed method approach [14], the qualitative (interviews & focus groups) and quantitative findings (RAS-DS and case note audit) are reported separately with convergence discussed in the interpretation of these results.

A focus group was undertaken with six staff members (participation rate 71%) from a range of disciplines (See Table 1 for staffing profile). A majority (82%, n = 5) had worked in the service for over a year. Individual interviews were carried out with two staff members, one of whom was unable to attend the scheduled focus group and one who wanted to provide additional thoughts following the focus group session.

All current (n = 6, participation rate 100%) and two previous consumers consented to participate in the study (mean age 30.5, SD 10.9; range 18–52). Two consumers consented to the quantitative data collection (n = 8 in total) but not the qualitative interviews (n = 6 in total). Participants' length of stay in the service is presented in Table 2. Consumer participants

**Table 1. Staffing profile of the CMHRS.**

| Budgeted staffing profile. * *Note not all positioned were filled at the time of data collection* | |
|---|---|
| Team Leader | 1 FTE |
| Occupational Therapist | 1 FTE |
| Social Worker | 1 FTE |
| Mental Health Clinician | 2 FTE |
| Psychologist | 0.8 |
| Nurse | 1 FTE |
| Support Worker | 5 FTE |
| Peer Support Worker | 0.7 FTE |
| Aboriginal Wellbeing worker | 0.5 FTE |
| Administrative Worker | 0.8 FTE |
| Consulting Psychiatrist | 0.5 FTE |
| Total budgeted staff | 14.3 FTE |
| *Total staff at time of data collection | 8.5 FTE |

**Table 2. Case note audit.**

| Data | Participant 1 | Participant 2 | Participant 3 | Participant 4 | Participant 5 | Participant 6 | *Participant 7 | Participant 8 |
|---|---|---|---|---|---|---|---|---|
| Age (years) | 18 | 21 | 27 | 29 | 39 | 25 | 52 | 33 |
| Length stay (Days) | 322 | 135 | 377 | 205 | 253 | 241 | 250 | 149 |
| Program activities | | | | | | | | |
| Therapeutic Groups | 137 | 101 | 145 | 32 | 18 | 6 | 68 | 30 |
| 1:1 Intervention | 477 | 200 | 484 | 58 | 91 | 34 | 55 | 60 |
| Clinical Care Review | 65 | 34 | 118 | 7 | 10 | 6 | 20 | 16 |
| Independent Activity | 63 | 33 | 55 | 8 | 40 | 4 | 8 | 22 |
| Total engagement | 742 | 368 | 802 | 106 | 160 | 50 | 151 | 13 |
| Phases of support | | | | | | | | |
| Phases of Support—pre | 1 | 2.25 | 1.3 | 1.5 | 1.4 | 1 | 1 | 1.5 |
| Phases of Support—post | 3.5 | 3.33 | 3.6 | 3.5 | 2.25 | 3.16 | 3.75 | 3.43 |
| RAS-DS scores | | | | | | | | |
| Raw RAS-DS scores pre | 98 | 81 | 91 | 85 | 104 | 95 | 123 | 145 |
| Raw RAS-DS scores post | 126 | 90 | 96 | 87 | 120 | 127 | 142 | 148 |

*Frequent Leave from the service.

Therapeutic Groups: Psychoeducational, gardening, meditation, exercise, cooking, transport, budgeting, creative, social. The table indicates the total documented number of groups attended.

1:1 Intervention: Rehabilitative intervention inclusive of planning and mental health support (facilitated with staff member). The table indicates the total documented number of 1:1 sessions attended.

Clinical Care Review: Clinical Review and planning, collaborative with multi-disciplinary team and consumer. The table indicates the total documented number of clinical care reviews.

Independent Activity: Use of free time and self-initiated activities. The table indicates the total documented free time and self-initiated activities.

Phases of Support: level of support required by consumers to participate in program activities. This ranges from 1 side by side support to initiate and complete tasks to 4 initiates and completes tasks independently. Pre = Average level of support documented in their initial Individual Rehabilitation Plan (IRP) across all program activities. Post = Average level of support required during their last Individual Rehabilitation Plan across all program activities.

(current and previous) were typical of the consumer profile of the service (See S1 File). The diagnosis of participants included schizophrenia, schizoaffective disorder, and severe depressive disorder.

The qualitative findings have been categorised in relation to the Context-Mechanism-Outcome configuration, recognising that these are inter-related; that is, CMHRS 'Context' influenced whether particular 'Mechanisms' operated to produce program 'Outcomes'. Within this configuration, thematic analysis identified nine primary themes, some of which comprised subthemes (total = 8) that highlighted nuances within the primary themes (See Fig 2). Participant quotes supporting identified primary themes and subthemes are denoted by: staff focus group (FG), individual interviews with staff/health professionals (HP), and consumer interviews (CP).

## Context

The variation between program participants was heavily influenced by the service context, including consumer, external stakeholder and staff interpretation and application of the CMHRS eligibility criteria.

**Appropriate referrals.** Staff expressed the importance of appropriate referrals being made to the CMHRS, to ensure delivery of the intended service to support consumer recovery. However, this became problematic when consumers with higher functional support needs, unstable housing, or very recent medication changes, were referred to the CMHRS.

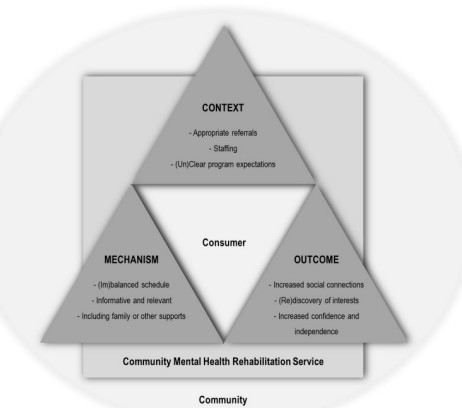

**Fig 2. Primary themes.**

FG001.6: That's hard because people can't even tie shoelaces and stuff like that.

FG001.1: Sometimes we'll get consumers who come to us, and those people will be sent to us for the wrong reasons, maybe because they don't have a house right there and then, and there will just be a shelter for them to stay in the house.

FG001.3: It's also not helpful sending them to us 24 hours after their medication has been changed.

*Risk.* The extent to which staff deemed a referral as appropriate, also had co-occurring implications in terms of risk, which particularly manifested from changed medication regimes and availability of appropriately qualified staff to support medication management.

FG001.1: We don't have an RN [Registered Nurse] that can do OBs [observations], we've had people on clozapine. . . we have to get OBs [observations] done by a nurse, we have to try and chase a nurse up to do OBs [observations] for them.

The sub-theme of 'risk' extends into the second primary theme, which focused on adequate staffing levels. This was identified by both staff and consumers as influencing the extent to which coordinated, supportive multi-disciplinary care, and responsiveness to needs was experienced within a service context. This, in turn, influenced the operationalization of particular mechanisms.

**Staffing.** CMHRS staff emphasised the service was 'under-staffed', and that this resulted in them having less time to spend one on one with consumers and to devote to developing group-based structures.

FG001.1: And because we're under-staffed, it's hard to spread ourselves for that one-on-one stuff. . .

FG001.3: . . . I think because there is no actual group structure a lot of the time, we're really struggling.

*Co-ordinated, supportive multidisciplinary care.* CMHRS staff highlighted the importance of a co-ordinated, multi-disciplinary team in terms of service provision during the consumer journey, particularly on admission and during comprehensive assessments.

FG001.4: . . .[we] include the clinical co-ordinator when we have an admission or have to do a comprehensive assessment, including mental state examination and risk assessment, but also work with the consumer to develop their individual health plan. Also work with the multidisciplinary team, liaise with internal and external organisations in order to meet the consumer need.

*Responsive to needs.* Even though staff reported under-staffing within the CMHRS context, this did not appear to impact consumers' experiences of feeling their needs could be responded to.

CP004: I've always got someone to talk to if I need to talk to someone. . . Just talk about all the issues I've got and all the problems I've got and try and help me go through it all and yeah.

This sentiment resonated with staff who indicated trying their best to accommodate consumer needs, however they also noted the challenge of creating group activities to meet a wide range of needs.

FG001.6: We've got residents who are saying they'd like to do something—we'll try and accommodate that.

FG001.2: I think some groups aren't fit for every client. They all have different needs. It's really hard to create a program because everyone has needs.

**(Un)Clear program expectations.** The third primary theme related to the extent to which program expectations were clear for both staff and consumers. Aspects relating to appropriate referrals, particularly prior to consumers' admission into the CMHRS, influenced their experience.

For consumers, there was variation in clarity regarding what to expect before, during, and after participating in CMHRS.

CP002: . . . maybe a fact sheet or something would be nice because I didn't get any information [on intake] except for the brief DTN [Digital Telehealth Network] (meeting).

CP005: It was good that they filled me in on how long it may take. They said it might take a few months to a year and, yeah, it did take that much. . . They gave me information on like what to bring, like what bedding and clothes and how much food to bring.

CP006: We had a meeting on discharge. What our goals were afterwards, and what we would do for when I get home. . ..

Confusion regarding the voluntary nature of the CMHRS was expressed by consumers, which was consistent with staff experiences.

CP001: It's supposed to be voluntary, that's what I didn't understand, and I wasn't allowed to leave. [The consultant medical practitioner] had me on order-thing [Community Treatment Order] and I wasn't allowed to leave and it's a voluntary place.

FG001.1 And sometimes they'll [consumer] say, 'but I don't want to be here [CMHRS]. I was told that I have to be here'

The following three subthemes highlight prominent aspects regarding reported consumer and staff expectations of the program.

**Goal setting.** Goal setting was identified as a clear expectation of the program by consumers. Many consumers indicated their CMHRS goals were long-term and predominantly related to paid employment or the process of 'getting a job'. However, one consumer detailed specific skills related to maintaining social connections and establishing independence with household management and community access.

> CP003: Do an aged care disability course . . . Just get a job again and be happy.

> CP002: I asked for help with my legal matters and I asked for help to re-establish communication with my parents and for help with the housing and the rest of it is part of the program you do here already, with the cooking and the budgeting and the transport training.

Staff referred to the importance of spending time with consumers to build rapport to support them with goal setting, linking to the importance of a supportive and responsive multidisciplinary context.

> FG001.1: Part of my role is to help create rapport with that person and help them through their goals, depending on the consumer and what their needs are.

*Support to navigate systems*. Consumers highlighted that the multidisciplinary team had assisted them to develop skills to navigate government services/systems which they would need to engage with following discharge.

> CP002: They've just been very helpful and helping me out with Centrelink and going to Housing SA. . .

*Transport*. Staff expressed concerns regarding providing transport, which they did not perceive to fit with the rehabilitative context of CMHRS and recovery-oriented practice. However, consumers indicated they found the organised transport helpful.

> HP002: I guess the focus has been a great deal of socialisation and clients and transporting them . . . some of these clients we already had catching buses and doing stuff independently. That stopped. . . .. that's not rehab. Rehab is when you go back out there in the community to homes or whatever, being able to find ways of getting yourself around. Getting yourself from A to B. Not having to rely on other people.

> CP005: . . . if we needed to be dropped off some where they would help us get there from place to place, so I found that helpful how they had transportation for us.

## Mechanism

Contextual themes influenced whether and which of the following three primary mechanism themes were operationalised.

**(Im)balanced schedule.** The service is designed to deliver a combination of structured one-to-one and group-based activities, as well as unstructured time for self-directed activity or 'free time' for consumers. The response was varied regarding the overall balance of structured and unstructured time from both staff and consumer perspectives. For some consumers, unstructured time was a positive experience to relax and socialise with others.

CP002: Yes, it's good, because we're not too overloaded with groups, because people like to socialise without having the staff around. . . And we all like to relax as well as going to groups, so I think it's a good balance.

Contrastingly, others perceived there was an imbalance with too much unstructured time increasing feelings of anxiety or boredom, and challenges relating to being away from their usual context and personal resources that enable participation in valued activity.

CP006: The free time I felt a bit unstructured . . . It made me feel anxious, waiting at home for the next group.

CP004: I don't know what there is to do really. Back home, you know, on a good day I'll take my boat out and go fishing. But can't do that here.

Staff expressed trying to overcome perceived schedule imbalances through supporting consumers to identify new or previously valued activities.

FG001.3: [Consumers] say, 'I'm bored'. . . You'll go through interest checks with them and you'll try and find new things. . . and then trying to reintroduce it to them almost again, or something similar.

Nested within this primary theme, was a subtheme relating to consumers motivation to participate.

*Motivation to participate*. Both consumers and staff described a point-based reward system (mechanism) designed to encourage consumer motivation, thereby increasing participation in structured and unstructured rehabilitative activities. One consumer indicated this motivated them to exercise, but there were implications in terms of fatigue, which impacted on participation in an unstructured activity they enjoyed.

CP002: Well, you get certain points for doing–it encourages you to join in activities and do things. You even get points for going to the gym, so I've been getting points for when I go to aquafit. I was going to the gym every morning, but I was getting really tired and I was falling asleep at ridiculous times, like seven o'clock while Neighbours was still on and I was missing out on my favourite shows. But I still go to aquafit.

Staff expressed concern regarding the intent of the motivational point-based reward system, offering that it could be problematic for facilitating intrinsic motivation towards activities that align with the service context regarding rehabilitation.

HP001: Before that was introduced [points-based reward point system], they would just go because it was something that benefited them to learn, and it's part of their rehab [rehabilitation]. . . But then that was brought in. . . some consumers, "Well, what's the point in me doing that if I'm not going to be rewarded?"

**Informative and relevant.** Consumers found the structured activities informative and relevant, especially the groups that focused on mental health conditions, information, and resources to support consumers with their recovery.

CP005: We had meetings about mental health, like had information about anxiety and depression and remembering things. We had some reading groups about that.

However, some consumers felt the skill-based activity regarding home maintenance was their least favourite group, with one consumer suggesting it could be shortened with additional take home information.

> CP002: Like I said I'd shorten maintenance–it doesn't have to go for half an hour–it can just go for 15 minutes . . . And maybe give us a form of what you do, so you've got some notes to take home, like how to clean the microwave, how to do the mop, how to do the oven. . .. so when we're out on our own we can look back on it.

The importance of staff creating structured group-based activities that are interesting and relevant resonated with consumer sentiment and the context theme related to staffing.

> HP001: . . . so long as it's of interest and relevance to them, yeah, the groups work well. Just making sure that they're structured for the different stages that different consumers are at.

The support provided by CMHRS staff was predominantly perceived by consumers as responsive and collaborative, which was consistent with staff member perceptions. However, not all consumers felt they had the opportunity to contribute to decision making, impacting the extent to which consumers felt informed and was relevant to their needs.

> FG001.4: Everything is about them. It's not about us. We just guided them and encourage them to get them to return—it's their rehab and we're here.

> CP006: I had enough opinion, in saying, because I could say I wanted to go home, I could, and if I didn't want to go to a group I didn't have to.

> CP005: Yeah, I didn't have much of a say, I just had to take the medication and I had to stay there for as long as it took to get the right dosage.

**Including family or other supports.** Many consumers reported the significant role their families or other supports played; particularly during admission to CMHRS which provided important opportunities for staff to discuss what to expect while participating in the CMHRS.

> CP001: . . .I got accepted in. Then we all sat down, me, Mum and some of the staff members and we just talked about everything and what's going to be happening in the program and they were really good with that sort of stuff.

Consumers also reported the importance of maintaining connections with family or other supports during the program. This was especially important for one consumer who would create opportunities during unstructured time to connect with family.

> CP001: So I just go out with family, go out for lunch with them or go down the beach and go to the cafe, go to [shopping centre], . . . I'm usually with family when I have my free days.

Another consumer reported the positive experience of staff engaging with family to give an update on their rehabilitative progress.

> CP002: Well [staff name] got on the phone first to my Mum and just told her how well I was doing and what she thought and Mum was all 'wow, wow, wow'. Are we talking about the same girl? Are you telling me the truth, are you for real?

From the staff perspective, the importance of liaising with family regarding goal setting and discharge planning was highlighted.

> HP001: . . . I think you have clear goals on—the family can come in and go, "Actually, we don't want this person to come back and live with us. We want them to live on their own." So at least then we know okay, we're not going to be surprised seven months down the track when we're ready to discharge.

## Outcome

The intended or unintended impact of contextual and mechanism components generate outcomes with three primary themes identified.

**Increased social connections.** The mechanism of a balanced schedule was reinforced by the context of staffing in some instances, which resulted in consumers building social connections during group-based activities or unstructured time.

> CP005: Well, yeah, we did [cooking] in a group. We all got involved, like we cleaned the dishes or one person might clean the dishes, one person might dry them, the other one might like the vegetables, yep.

> CP004: I do things here, just connect with other people and have coffee with people and yeah. That sort of puts a smile on my face.

**(Re) discovery of interests.** Enabling contextual factors, such as staffing and transport, as well as mechanisms to promote participation, enabled participation in activities of interest which contributed to recovering consumers' health and wellbeing.

> CP002: I started reading again. They take us to the op shops too on Saturdays, and I've been getting books—just from the op shops, and quite good books fairly cheap–I just picked up a John Grisham one for 50 cents.

However, not all consumers felt there was enough opportunity to engage in valued activity, which was impacted by lack of access and resources due to the rural location of the program. It is important to note staff were perceived as supportive by consumers in their attempts to assist them with engaging in valued activity.

> CP003: . . . I like fishing and things like that, footy and cricket. . . I like to go watch it (footy). . . They're [staff] going to try and get support to go with you to watch the games but it never happens.

**Increased confidence and independence.** Contextual aspects of staff who were responsive to the needs of consumers and mechanisms, such as structured activities, increased consumer confidence and independence in social situations, as well as capacity to engage in and implement strategies that support recovery, health, and wellbeing.

> CP001: I just really enjoy cooking and that sort of stuff, it just—because I was back at home. . . this [cooking group] has just gotten me out of my shell . . . I'm really shy, that's why I don't make much eye contact because I get really shy, but I don't know, it was just really, really good.

CP006: I can go to more places with the strategies that they in placed—not in placed, that they gave me. And I used the strategies for my recovery, and the more I used it the more confident I got.

Staff also reported consumers increasing confidence and independence through structured activities in supportive environments.

FG001.2: Let's just bring them [consumer] and maybe they can bring their knowledge in, and then they start almost co-facilitating it with you. And it gives them self-esteem and the other guys are going, wow, this is awesome. If that person can do it, I can.

## Quantitative results

**RAS-DS.** Qualitative program outcomes were supported by the RAS-DS results (see Table 3). There were significant positive increases in RAS-DS total scores from a mean (SD) of 102.8 (21.4) at admission to 117 (23.4) at discharge (p = .010). A similar pattern was noted for three of the domain scores: 'Doing Things I Value' (admission 15.9 (3.4), discharge 18.5 (3.3), p = .017), 'Looking Forward' (admission 49 (11.2), discharge 56.1 (12.6), p = .027), and 'Mastering My Illness' (admission 20.5 (4.8), discharge 23.5 (7), p = .020). Mean scores for the 'Connecting and Belonging' domain increased from admission 17.4(4.9) to discharge 19.6(6) but was not statistically significant (p = .101).

**Case note audit.** Table 2 summarises the case note audit data and shows varying level of interaction with service activities by consumers. The level of support consumers required (for activities) decreased significantly (see Table 3) during their stay at CMHRS (admission 1.4 (0.42), discharge 3.3 (0.47); p = .012). Consumers generally progressed from requiring side by side support to initiate and complete tasks successfully, to reaching a stage where they were able to initiate and complete tasks independently or with stand by assistance.

## Discussion

This mixed methods realist evaluation of the CMHRS was designed to identify consumer and staff perceptions of the critical aspects influencing service delivery and the impact of the service

**Table 3. Pre-post statistical analysis of RAS-DS & level of support.**

| Normally distributed data | | | | | | | | |
|---|---|---|---|---|---|---|---|---|
| Outcome | Pre-test mean (SD) | Post-test mean (SD) | Pre-test median (IQR) | Post-test median (IQR) | Paired t-test | | Effect Size | |
| | | | | | t | Significance 2-tailed | Cohens d | 95% CI |
| RAS-DS total score | 102.8 (21.4) | 117 (23.4) | 96.5 (31.8) | 123 (46.8) | -3.5 | .010 | 0.63 | -0.79–2.1 |
| RAS-DS Domain scores | | | | | | | | |
| Looking Forward | 49 (11.2) | 56.1 (12.6) | 46 (17.5) | 59.5 (23.5) | -2.8 | .027 | 0.50 | -0.82–2.0 |
| Mastering my illness | 20.5 (4.8) | 22.8 (3.9) | 19.5 (8.3) | 23.5 (7) | -3.0 | .020 | 0.53 | -0.88–1.9 |
| Connecting and belonging | 17.4 (4.9) | 19.6 (6) | 18 (8.8) | 19.5 (10.5) | -1.9 | .101 | 0.40 | -0.99–1.8 |
| Non-normally distributed data | | | | | | | | |
| Outcome | Pre-test mean (SD) | Post-test mean (SD) | Pre-test median (IQR) | Post-test median (IQR) | Wilcoxon Signed Ranks Test | | Effect Size | |
| | | | | | Z | Significance 2-tailed | Cohens d | 95% CI |
| RAS-DS Domain 'Doing Thing's, I Value' | 15.9 (3.4) | 18.5 (3.3) | 15 (2) | 17.5 (6) | -2.4 | .017 | 0.78 | -0.66–2.2 |
| Level of Support required | 1.4 (.42) | 3.3 (.47) | 1.4 (.50) | 3.5 (.37) | -2.5 | .012 | 4.3 | 1.8–6.7 |

on consumer outcomes. To assist with exploring these concepts, a preliminary program theory was developed (see Fig 1), using a Context-Mechanism-Outcome (CMO) configuration [7]. The study findings support the content (critical aspects under the CMO configuration) of the preliminary program theory; however, some areas for improvement that could further enhance the effectiveness of the service were identified. These included appropriate referrals and intake of consumers into the service (context); adequate staffing (context); (un)clear program expectations (context); and refinement of group programs and unstructured time (mechanisms). Consumers indicated they had experienced increased social connectedness and had an opportunity to pursue vocational, domestic, and leisure goals (outcomes), which is supported by RAS-DS results. Further individual tailoring of support is important in facilitating independence in the community.

## Context

One of the main concerns raised by staff was the appropriateness of referrals accepted by the service. Staff indicated they felt some consumers were not ready for rehabilitation (e.g. needed support with self-care or had recent medication changes) or did not want to attend the voluntary service (this was also supported by consumer comments). This can impact on appropriate use of resources (e.g. staff time) and result in staff dealing with crisis situations rather than assisting consumers with identifying and working on long-term rehabilitation goals. Previous research has highlighted that, despite attempts in Australia and across the globe to facilitate partnership and integration of mental health services (e.g. South Australia's Stepped Model of Care), fragmentation is still experienced due to system complexity [15]. Therefore, further exploration of factors which may help to build communication, collaboration, and partnership between the service (management and staff), consumers, and referrers (at an organisation and individual level) is required. Revision of referral processes including policies, information, and education for consumers/families/referrers about the service (particularly in relation to its rehabilitation focus), referral documentation and procedures is needed.

A multidisciplinary team is an essential component of a rehabilitation service, as they bring discipline specific expertise that assists with meeting a range of consumer rehabilitative needs. Having adequate staffing to provide intensive and responsive rehabilitative intervention underpins good practice [16]. Consumers generally reported that staff were available and responsive to their needs. Staff also highlighted the importance of an individually tailored rehabilitation approach, however, expressed difficulties achieving this due to staff shortages. This impacted on their ability to spend one on one time with consumers and to develop group programs and structures to address consumer goals. The service has experienced chronic issues with recruitment and retention since its inception, this has had a perceived impact on its ability to function from a rehabilitation and recovery perspective. High staff turnover does not allow for growth of knowledge and skills within the service or refinement and development of interventions and supports. Staff retention is a common issue in rural mental health services in Australia, with the following factors contributing to this—heavy workload, complexity of consumer needs, and lack of opportunity to develop a discipline specific identity and skills [17].

Several strategies have been established by the Australian Federal and State governments to attract a greater rural health workforce, including funding rural university clinical placements, setting quotas for rural background students to attend university health programs, and financial incentives and supports (e.g. increased professional development opportunities) [17, 18]. However, the need to explore personal, career, social, and community factors, as well as service and community specific supports, to build on these strategies could be implemented at the

local level for the CMHRS. For example, assisting new staff to make connections to support their personal and social goals [18, 19], and providing regular ongoing training to assist staff with applying rehabilitation recovery principles with CMHRS consumers.

## Mechanism

The balance between structured and unstructured time was mentioned by a number of consumers and staff with opinions varying. This was supported by the case note audit results which showed a varied level of consumer interaction with program activities. Some consumers indicated they enjoyed the unstructured time as it enabled them to socialise with other consumers in a relaxed manner, whereas others found this challenging commenting there was not a lot to do. Staff expressed similar views indicating they would try to spend one on one time with consumers to help them to identify ways of decreasing boredom (via interest checklists).

On the whole, consumers commented positively on the group programs (which focussed on the development of independent living skills, including. cooking, budgeting, and managing health), however staff and consumers did not feel group programs were always meeting consumer needs or had enough of a rehabilitation focus. The value of therapeutic groups is well established in the literature [20], not just for the content presented but the indirect effects such as social interaction, altruism, sharing of stories, roles, and membership [20]. To address anxieties for some consumers regarding unstructured time, adding more groups to the CMHRS program would allow for greater choice in how time is spent. Similarly, looking at alternative ways to assist consumers to identify and explore different occupations would be beneficial, for example come and try activity sessions within the CMHRS and the larger community. One consumer reported he enjoyed going fishing but was unable to pursue this, despite the CMHRS being located in a seaside town. Another consumer expressed a desire to attend football games which staff had indicated that they would follow up, however this had not happened. Staff discussed issues regarding over reliance on service transport, rather than consumers developing the skills to use public transport. These comments indicate that some additional supports and skill development may enable consumers to effectively use their time to achieve goals. Consumers come from rural towns across South Australia and the services/activities (such as public transport) may vary greatly. These factors need to be considered when supporting consumers, and in the design and development of rehabilitation activities.

## Outcomes

Despite some of the challenges that have been discussed in relation to the service context and mechanisms, on the whole, consumers commented positively about the service. Particularly in relation to their perceived recovery outcomes. This was evident in the evaluation themes–'increased social connections', '(re) discovery of interests' and 'increased confidence and independence'. This was supported by the case note audit, which showed decreased levels of support were required for program activities as consumers progressed through the service. RAS-DS pre-post findings also indicated significant improvements in total scores and domain scores for 'Doing things I Value', 'Looking Forward' and 'Mastering my Illness'. Interestingly, the pre-post scores for 'Connecting and Belonging' domain was not statistically significant. While consumers discussed feeling a sense of social connection and belonging with other consumers in the service, there was less reference to connecting with the wider community. This is a potential unintended consequence of the clustered housing model.

However, it is important to note that the CMHRS is the only residential mental health rehabilitation service in regional and remote South Australia. Consumers often need to leave their town of origin and travel to a new community, away from their established social networks

and supports, to access the service. This may lead to difficulties in maintaining relationships or re-establishing connections on discharge.

A systematic review by Webber & Fendt-Newin [21] found mental health services rarely include sufficient interventions to assist consumers to improve or build their social networks. Potential interventions include, social skills training, supported community engagement, and employment support. In addition to exploring and reviewing these opportunities, the CMHRS should review discharge processes with a graded approach for consumers to reconnect with family, friends, and community (town of origin) through planned leave.

## Strengths & limitations

The realist evaluation approach has enabled exploration of a preliminary program theory and the development of a middle range theory, identifying areas for service refinement. A mixed method approach utilising the perspectives of a diverse range of staff and consumers, and tri-angulation of qualitative data with RAS-DS scores and case note audits, has strengthened study rigour.

As with most service specific evaluations, generalisation of findings may not be possible. The sample size for this study was small and reflective of the nature of the service, including staffing at the time. While past service users were included, they could only be recruited if they had a current case coordinator also limiting sample numbers. It is believed that saturation was reached with the study sample (staff and consumers) based on limited new information being provided towards the end of data collection.

It is acknowledged that the perceptions of all potential key stakeholders, such as consumers' families, referrers, and post-discharge case coordinators, have not been captured (as they were not part of the study inclusion criteria) and could be the focus of future research. Further exploration of consumers' mental health history and the impact on service engagement and RAS-DS scores, may provide additional information regarding suitability for different consumer groups. Longer-term outcomes for consumers (6 to 12 months post-discharge) were not measured and this will be an important addition to ongoing service evaluations. Finally, further exploration of staff support needs and collaborative approaches for designing program activities/interventions would also be beneficial.

## Conclusion

This evaluation used a realist evaluation approach to explore the CMHRS in terms of context, mechanisms, and outcomes. While the service was seen to have positive outcomes for consumer recovery, the evaluation identified areas where the service could be strengthened, including appropriate service referrals; strategies for improving staff retention; providing increased choice and support around how consumers use their time; ensuring that program activities are rehabilitation and recovery focussed; and increasing ways consumers can connect with the wider community. Gaps where additional information is required to develop an in-depth understanding of service operations and outcomes were also identified, such as hearing perspectives of all key stakeholders, and looking at long-term consumer recovery outcomes 6–12 months post-discharge.

## Supporting information

**S1 File. Service eligibility and consumer profile.**
(DOCX)

**S2 File. Interview and focus group guide.**
(DOCX)

## Author Contributions

**Conceptualization:** A. Leet, S. Dennis, S. Walsh, J. Kernot.

**Data curation:** A. Leet, S. Dennis, J. Muller.

**Formal analysis:** A. Leet, S. Dennis, J. Muller, S. Walsh, J. Kernot.

**Funding acquisition:** A. Leet, S. Dennis, S. Walsh, J. Kernot.

**Investigation:** A. Leet, S. Dennis, J. Muller, S. Walsh, J. Kernot.

**Methodology:** A. Leet, S. Dennis, J. Muller, S. Walsh, J. Kernot.

**Project administration:** A. Leet, S. Dennis, J. Muller, S. Walsh, J. Kernot.

**Resources:** A. Leet, S. Dennis, J. Muller, S. Walsh, J. Kernot.

**Writing – original draft:** A. Leet, S. Dennis, J. Muller, S. Walsh, J. Kernot.

**Writing – review & editing:** A. Leet, S. Dennis, J. Muller, S. Walsh, H. Bowen-Salter, J. Kernot.

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
