## [Decision Letter · Decision Letter 0]

4 May 2021

PONE-D-21-02032

Demonstrating the processes and outcomes of a rural Community Mental Health Rehabilitation service: A realist evaluation

PLOS ONE

Dear Dr. Jocelyn Kernot,

Thank you for submitting your manuscript to PLOS ONE. After careful consideration, we feel that it has merit but does not fully meet PLOS ONE’s publication criteria as it currently stands. Therefore, we invite you to submit a revised version of the manuscript that addresses the points raised during the review process.

Your manuscript focuses an important topic area. Please address all comments from both reviewers as well as addressing potential confidentiality issues in Table 1. 

We look forward to receiving your revised manuscript.

Kind regards,

Nelly Oelke

Academic Editor

PLOS ONE

Additional Editor Comments:

This manuscript addresses an important topic.

Both reviewers have provided detailed feedback that authors will need to address.

In addition, Table 1 includes ages of individual participants. This information is potentially identifiable and should not be included as presented. Please number participants as 1, 2, etc. or something like that, and remove the ages. Present age range, average, median, etc. in the text.

Journal Requirements:

2. We note that you have included blacked out text in the methods section of your manuscript. As PLOS ONE is not a double blind peer review journal can you please insert the missing text so all details can be read in full.

Reviewers' comments:

Reviewer's Responses to Questions

**Comments to the Author**

1. Is the manuscript technically sound, and do the data support the conclusions?

Reviewer #1: Yes

Reviewer #2: Yes

2. Has the statistical analysis been performed appropriately and rigorously? 

Reviewer #1: I Don't Know

Reviewer #2: No

3. Have the authors made all data underlying the findings in their manuscript fully available?

Reviewer #1: No

Reviewer #2: No

4. Is the manuscript presented in an intelligible fashion and written in standard English?

Reviewer #1: Yes

Reviewer #2: Yes

5. Review Comments to the Author

Reviewer #1: Realist evaluation of a mental health service – Reviewer comments

Thank you to the author team and the MH service for enabling a realist evaluation of the service. It is important to build and show the evidence of how rural mental health services work, their achievements and challenges in a systematic but pragmatic way.

There are valuable lessons in referral appropriateness and program expectations, contending with workforce turnover, balancing support with skill-building for future independence across several domains (social connectedness, domestic and vocational skills).

I think the manuscript could be published if the comments below can be addressed.

Line 57 the CMHRS is referred to as a 10-bed mental health residential program, however, in line 581 it is referred to as clustered housing. Could the authors please detail the program a little more clearly so that the reader has a context to put it in? People living 1-3 in clustered housing sounds very different to a 10-bed mental health residential program (which still manages to conjure up a ward image in my mind!).

Line 78 - Could you also elaborate on the stepped model in this context? It is indicated in the figure to an extent, but this is introduced later – some clarification in the text would be beneficial.

Line 115 – Purposive sample - could you indicate how many people were invited, who accepted and thus a participation rate? (could be reported at Line 175) With your note of staff turnover – was there a mix of established and newer staff? I am wondering about their ability to judge the care model if they were relatively new.

Line 119 – a single focus group of an hour is short, especially with two additional interviews (assume one 30 minutes and one 45 minutes) – can the authors comment on this? Was saturation reached or is this a limitation of the research, if so please state this.

Lines 124-25 – regarding piloting and change to interview guide, currently vague – could you elaborate some more?

Line 132 – can you elaborate on the practical application of steps of thematic analysis – how many authors participated? How was coding performed (more than one author)? How were themes reviewed and defined? – were the broader authorship group brought in? If any, how were differing opinions settled? Cannot judge the rigour of this approach without more detail.

Line 149 – was the consumer data analysed completely separately? Were the themes brought together at some point? It would make sense that this was the case – can the authors elaborate?

Line 171 – this appears to be a description of the analytic method – I suggest it be moved into the methods section.

Line 175 – I would suggest that the sample is small, this should be listed in limitations, since it is a 10-bed program could you comment on the sample relative to that. Perhaps there could be some discussion on whether themes were saturated within the sample, thereby suggestive of sufficiency or not. The fact that the service is tailored and meets consumers’ needs – variable treatment, support and length of stay, would suggest that this may not be the case (my speculation).

Line 186 – would the authors consider including a table of themes and sub-themes as a supplementary table? I was interested to see the nuance – if it is what you describe in the rest of the results, say so.

Line 188 – was the consumer sample representative of the range of consumers who used the service?

Line 188 – table 1, participant 7 has a subscript 17 – there is no footnote – is it the *? Please clarify.

Line 198 – ‘appropriate referrals’ – what do you mean? Could you provide a clearer explanation? E.g. State what a ready consumer looks like perhaps. Lines 82-83 indicate eligibility as having ‘a serious mental illness and identified rehabilitative needs and goals’ – are there exclusion criteria, or more explicit inclusion criteria? Is this a point for the discussion?

Line 203 – pedantic, but could you identify somewhere the schema for the quotes (clearly FG means focus group, HP health professional, CP consumer participant… but it would be good for clarity)

Lines 226-228 – could you elaborate on what you mean? Especially as it is unclear what ‘no actual group structure’ meant in line 231. I think it is just a bit vague and not cutting through – do you mean that due to understaffing, there was a lack of staff time to devote to developing group-based structures nor one-to-one time for consumers.?

Lines 274-276 – does this quote merit discussion also – in terms of the voluntary nature of the program – if someone is on an order, would that render them ineligible for a rehab transition to community service? Have they been left there for some other logistical convenience – no beds in acute for example?

Line 278 – I don’t understand this quote – is there context to the statement – what is the ‘this’ in ‘I was told I had to be here to get to this? Sorry?’

Lines 300-302 – awkward sentence – presumably an opportunity for support and the development of skills to navigate the complex systems of government support that consumers would need to engage with following discharge?

Line 307 – suggest that after CMHRS include ‘and recovery-oriented practice’.

Lines 346-348 – not sure if the quote backs up the assertion – was the imbalance in the short term here? The fact that CP002 stills goes to aquafit may indicate a prioritising of supportive activities that might cause short term fatigue.

Lines 388-390 – is there another quote to support this? The following two quotes (Lines 391-394 support the first part of the paragraph, but not the second).

Line 470 – I got a mean of 103(21) and 117(23) – could the authors verify their calculations and the stated figures in Table 1? (this would include the sub-domain scores too please).

Lines 520-522 – the results mention understaffing; however staff shortages, recruitment and retention issues are not explicitly discussed – my impression had been that perhaps the higher needs consumers were causing increased demand on the staff, thereby reducing hours available to the intended work of the program. Can these aspects be introduced and elaborated on in the results first – it is an important discussion point.

Lines 533, 537 – Ref 18 – could authors see newer refs by the same author that may be more appropriate - https://pubmed.ncbi.nlm.nih.gov/32295246/ and https://www.mdpi.com/1660-4601/17/8/2698

Lines 564-567 – I can see the tension between providing support and empowering independent use of skills (public transport on their own) – is there a reflection to be had for the rural nature of the CMHRS location and the likely relatively poor access to public transport, also for the relevance compared to what it will look like when they move on discharge (thinking specific vs general skills to navigate public transport in different places)?

Reviewer #2: This manuscript describes a research study designed to evaluate a community-based residential mental health rehabilitation program. This treatment program is intended to enhance accessibility to rehabilitative mental health services in rural communities within Southern Australia. As acknowledged by the authors, the limited access to specialist mental health programs in rural communities is a global issue. As such, the research addresses a very important and pertinent issue. Moreover, the described research suggests that the program helps to address the gaps in services accessible by adults with mental health concerns who live in a rural community. They also present evidence-based recommendations for changes that may serve to promote even greater effectiveness of the program.

The research was framed as a realist evaluation, which considers the inter-relationships between contextual variables (e.g., eligibility criteria for the program, staffing), mechanisms (e.g., components of services offered), and outcomes (e.g., the impacts, intended and otherwise, of the program). This theoretical approach is appropriate and promotes a more comprehensive consideration of variables that may be influencing the effectiveness of the program. Importantly, participants included both staff members and consumers (current and recently discharged). Moreover, the researchers highlighted the convergence and discrepancies of perspective. Another strength of the study was the use of mixed methods, including individual interviews (consumers and staff), and a focus group with staff members. Quantitative methods included a pre- /post-comparison of scores on the self-report questionnaire, Recovery Assessment Scale – Domains and Stages (RAS-DS; completed by consumers), and an audit review of case notes of participating consumers.

The theoretical framework, design, and methods were appropriate to the research questions. Moreover, the results are generally congruent with the authors’ claims. However, as described below, there were some concerns regarding the analysis of the quantitative data (or at least how the results were presented). In addition, inclusion of more detailed information about the participants, and perhaps, the program would help to strengthen the data analysis and, ultimately ,the interpretations of the results and arising recommendations. This information may also allow for further consideration of the program outcomes and recommendations for changes.

The recommended revisions are relatively minor (that is, they should be easily achieved); however, they will substantively improve the quality of the manuscript.

Concerns:

1. While the authors acknowledge the likely limits of generalizability of the results, additional information about the program and participants (both staff and consumers) would help in this regard. For example, the only mention of who the program is intended for is on p. 5 (line 40), when the authors indicate that individuals referred to the program “generally have a serious mental illness and identified rehabilitative needs and goals”. These descriptors are quite vague. Additional information would be helpful. For example, the definition of “serious mental illness” can vary tremendously. For some, this refers strictly to individuals with psychotic symptoms. Others define it in regards to the level of distress and/or impairment associated with the symptoms.

No information about the mental health history of the consumer participants was presented. While this may have been intended to help protect the confidentiality of these participants, it seems that some information could and should be provided. This would help give the reader a better understanding of the program and the identified strengths and weaknesses of it. Inspection of Table 1 suggests that the impact of the program varied substantially across participants. This seems to be the case both in relation to the RAS-DS data and levels of support required by consumers at the beginning and end of their stay. Is it possible that the impact of the program varied in relation to mental health history of the consumers? The variability in outcomes was not addressed by the researchers. While the small sample size limits the ability to reach any strong conclusions in this regard, the possibility merits consideration.

In a similar vein, very little information was provided about the staff participants. Again, it is stated in the introduction (p. 5, line 41) that the team is multi-disciplinary. However, the different professional disciplines included is never described. Nor is it known if the staff who participated in the research represented the breadth of the professions and staff roles. This is pertinent as it speaks to the representativeness of the views expressed by staff members, which may strengthen or weaken some of the conclusions.

2. Overall, the statistics were appropriate. Nonetheless, there were concerns about the

statistical analysis of the RAS-DS data. On p. 9, the authors indicated that they analyzed these data using the “Wisconsin paired-signed rank test”. Presumably, they meant the “Wilcoxon paired-signed rank test.” The rationale for use of this statistical procedure was not stated. However, this test is typically used when the distribution of scores deviates from a normal distribution. No information was provided about the shape or skewness of the distribution. If the distribution did, in fact, deviate from normal (which is likely given the small sample size), then the use of the mean scores (see pp. 24-25) is questionable. Generally, the median is used to summarize the data. In addition, the presentation of the results of these analyses was incomplete. While the authors indicated the associated significance levels, it would be helpful if they also stated the Z (or T) scores and an estimate of the effect size (typically, determined using rank-biserial correlations). Also, please clarify if one-tailed or two-tailed tests of significance were used. While a case could, perhaps, be made for one-tailed tests, two-tailed tests may be more appropriate since the authors are interested in what has been effective and what hasn’t. The directionality of the tests of significance should be specified and the rationale for this decision provided. Also, it was unclear why the quantitative information obtained from the audit case review was not analyzed in a similar manner. In particular, it seems that the pre- /post- level of support required could be analyzed.

3. As indicated in the response to the question concerning the quality of writing, in the main, it is written in standard English. However, the paper would benefit from careful proof-reading or use of a manuscript editing service. Sentences were often very long (e.g., 5+ lines in length) and awkwardly structured such that the writing distracted from the content of the manuscript. Revisions regarding appropriate use of the possessive and punctuation would be very helpful.

There were also some problems with the formatting of Table 1 and Figure 1. Despite several attempts to download Table 1, I was unable to download a copy that fit within the margins. Specifically, the left-hand column was partially cut-off. Please check the formatting. In a similar fashion, the resolution of Figure 1 was poor. Moreover, the size of the font was very small, making it very difficult to read.

6. PLOS authors have the option to publish the peer review history of their article (what does this mean?). If published, this will include your full peer review and any attached files.

Reviewer #1: **Yes: **Dr Hazel Dalton

Reviewer #2: No

---

## [Author Response · Author response to Decision Letter 0]

27 Aug 2021

Dear Editor,

Thank you very much for the opportunity to revise our manuscript PONE-D-21-02032

‘Demonstrating the processes and outcomes of a rural Community Mental Health Rehabilitation service: A realist evaluation’. We very much appreciate the reviewers’ time and and effort and feel that they have assisted us to strengthen this article. A response to each of the reviewers’ comments are provided below.

Editors Comments Response to Editor

1. Table 1 includes ages of individual participants. This information is potentially identifiable and should not be included as presented. Please number participants as 1,2 etc or something like that and remove the ages. Present age range, average, median, etc in the text Thank you, the ages have been deleted from the table as suggested and the mean age and age range included in the text 

All current (n=6, participation rate 100%) and two previous consumers consented to participate in the study (mean age 30.5, SD 10.9; range 18-52). 

2. Please ensure your article meets the PLOS ONE style requirements, including those for file naming as per the PLOS ONE style templates Thank you this has now been addressed

3. We note that you have included blacked out text in the methods section of your manuscript. As PLOS ONE in not a double blind peer review journal can you please insert the missing text so details can be read in full Thank you, this has been altered as suggested

4. We note that you have indicated that data from this study are available on request. PLOS ONE only allow data to be available upon request if there are legal or ethical restrictions on sharing data publicly We have read the information in more detail and we do feel that we have provided the ‘sufficient anonymized data necessary to replicate study findings’ within the manuscript itself. 

Quantitative data

For the quantitative data we have provided raw pre-post scores for the main outcomes (RAS-DS total scores and phases of support) for each participant in Table 2 as well as results of the statistical analysis in Table 3. We have also provided raw data (no. of sessions attended for each participant) for individual program activities and included totals (raw) to demonstrate levels of engagement. 

Qualitative data

For the qualitative data all themes and subthemes include quotes from participants to support findings. Figure 2 details the primary themes showing how they interconnect/relate to each other. To provide greater clarity regarding data collection procedure the semi-structured interview and focus group guides have been included in Supplementary Information 2

Reviewer 1 Comments Response to Reviewer 1

1. Line 57 the CMHRS is referred to as a 10-bed mental health residential program, however, in line 581 it is referred to as clustered housing. Could the authors please detail the program a little more clearly so that the reader has a context to put it in? People living 1-3 in clustered housing sounds very different to a 10-bed mental health residential program (which still manages to conjure up a ward image in my mind!). Thank you for this comment we have now changed this to 

This paper outlines a realist evaluation of a rural Community Mental Health Rehabilitation Service (CMHRS), a clustered housing (1-3 people per house)residential program offering rehabilitative mental health support to rural South Australians. The maximum number of consumers that can be in the program at one time is 10.

2. Line 78 - Could you also elaborate on the stepped model in this context? It is indicated in the figure to an extent, but this is introduced later – some clarification in the text would be beneficial. Shaun & Adrian Thank you, the following text has been added 

The CMHRS is part of the SA Health stepped model of care, which provides graduated/tiered levels of care including secure care, acute care, intermediate/sub-acute care, rehabilitation and supported accommodation. The CMHRS provides supported accommodation which aims to assist consumers to achieve and enhance independent living skills [9].

3. Line 115 – Purposive sample - could you indicate how many people were invited, who accepted and thus a participation rate? (could be reported at Line 175) With your note of staff turnover – was there a mix of established and newer staff? I am wondering about their ability to judge the care model if they were relatively new. 

Staff

There was a mix of established and newer staff who participated in the study, with a participation rate of 71%. To make this clear the following has been added to the results section 

A focus group was undertaken with six staff members (participation rate 71%) from a range of disciplines (See Table 1 for staffing profile). A majority (83%, n=5) had worked in the service for over a year (range 2.5 months to 4 years). Individual interviews were carried out with two staff members, one of whom was unable to attend the scheduled focus group (service length 4 months) and one who wanted to provide additional thoughts following the focus group session (service length 4 years). 

Consumers 

All current consumers of the CMHRS participated in the study 100%. It is difficult to calculate a participation rate for previous consumers as invites were sent via their case manager and it was difficult to keep track of the number of invitations sent out. The following has been added to the text 

All current (n=6, participation rate 100%) and two previous consumers consented to participate in the study (mean age 30.5, SD 10.9; range 18-52). Two consumers consented to the quantitative data collection (n=8 in total) but not the qualitative interviews (n=6 in total). Participants’ length of stay in the service is presented in Table 2. Consumer participants (current and previous) were typical of the consumer profile of the service (See SI 1). The diagnosis of participants included schizophrenia (n=6), schizoaffective disorder (n=1), and severe depressive disorder (n=1).

4. Line 119 – a single focus group of an hour is short, especially with two additional interviews (assume one 30 minutes and one 45 minutes) – can the authors comment on this? Was saturation reached or is this a limitation of the research, if so, please state this. The focus group duration (1 hour) was reasonably short and this could have been related to group dynamics. The two individual interviews did build on information provided during the focus group with one participant (who also attended the focus group) requesting an individual interview to expand on their answers. The other individual staff interview was with a staff member who could not attend the focus group

It is felt that saturation was achieved as staff reported they had no additional information to add at completion of the focus groups and interviews. A comment regarding saturation has now been added to the discussion 

The sample size for this study was small and reflective of the nature of the service, including staffing at the time. While past service users were included, they could only be recruited if they had a current case manager also limiting sample numbers. It is believed that saturation was reached with the study sample (staff and consumers) based on limited new information being provided towards the end of data collection.

5. Lines 124-25 – regarding piloting and change to interview guide, currently vague – could you elaborate some more? Thank you, the following has now been added to the text 

The interview guide (SI 2) was piloted with two health staff who were not involved with the CMHRS. Minor amendments were made based on the feedback received. These were primarily regarding wording and flow of the questions.

6. Line 132 – can you elaborate on the practical application of steps of thematic analysis – how many authors participated? How was coding performed (more than one author)? How were themes reviewed and defined? – were the broader authorship group brought in? If any, how were differing opinions settled? Cannot judge the rigour of this approach without more detail. Thank you, we have added further details to provide clarity about the practical application of the steps as detailed below 

The process of thematic analysis was guided by Braun and Clarke’s [11] six phases of thematic analysis: 1) familiarisation with the data through detailed reading of the transcripts (JM & JK); 2) generating codes using NVivo software (JM & JK) ; 3) searching for themes (JM & JK); 4) reviewing and 5) defining themes through meetings and discussion with the research team (all authors) and 6) producing a report (all authors). Differences in opinion at all stages were openly discussed (between team members involved) during face to face meetings and resolved through consensus)

7. Line 149 – was the consumer data analysed completely separately? Were the themes brought together at some point? It would make sense that this was the case – can the authors elaborate? The following explanation has been provided 

The consumer and staff data were analysed separately initially (Braun & Clarke phases 1-3) and then finding brought together when the authors were reviewing and defining the themes (Braun & Clarke phases 4-6). Findings were categorised and reported using the realist evaluation Context-Mechanism-Outcome configuration.

8. Line 171 – this appears to be a description of the analytic method – I suggest it be moved into the methods section. Thank you for this suggestion. We have included this information at the beginning of the results section to provide clarity as to how the results are reported and then combined (sign posting for the reader). That is that the qualitative findings are presented first and then the quantitative findings with relationships/convergence of results addressed in the discussion. We are reluctant to change this as we feel it could impact on clarity

9. I would suggest that the sample is small, this should be listed in limitations, since it is a 10-bed program could you comment on the sample relative to that. Perhaps there could be some discussion on whether themes were saturated within the sample, thereby suggestive of sufficiency or not. The fact that the service is tailored and meets consumers’ needs – variable treatment, support and length of stay, would suggest that this may not be the case (my speculation). As suggested, we have included the small sample size as a limitation in the discussion section 

The sample size for this study was small and reflective of the nature of the service, including staffing at the time. While past service users were included, they could only be recruited if they had a current case manager also limiting sample numbers. It is believed that saturation was reached with the study sample (staff and consumers) based on limited new information being provided towards the end of data collection.

10. Line 186 – would the authors consider including a table of themes and sub-themes as a supplementary table? I was interested to see the nuance – if it is what you describe in the rest of the results, say so. Instead of a table we have included a figure (Fig 2) which summarises the themes and subthemes

11. Line 188 – was the consumer sample representative of the range of consumers who used the service? Thank you we have added Supplementary Information (SI 1) to our paper which gives further details about eligibility for the service and a consumer profile. The following has been added 

All current (n=6, participation rate 100%) and two previous consumers consented to participate in the study (mean age 30.5, SD 10.9; range 18-52). Two consumers consented to the quantitative data collection (n=8 in total) but not the qualitative interviews (n=6 in total). Participants’ length of stay in the service is presented in Table 2. Consumer participants (current and previous) were typical of the consumer profile of the service (See SI 1).

12. Line 188 – table 1, participant 7 has a subscript 17 – there is no footnote – is it the *? Please clarify. For some reason the foot note was deleted when the table was uploaded. Yes, you are correct that it relates to * noting that the participant had frequent leave from the service

13 Line 198 – ‘appropriate referrals’ – what do you mean? Could you provide a clearer explanation? E.g. State what a ready consumer looks like perhaps. Lines 82-83 indicate eligibility as having ‘a serious mental illness and identified rehabilitative needs and goals’ – are there exclusion criteria, or more explicit inclusion criteria? Is this a point for the discussion? 

 To address this, we have provided Supplementary Information (SI 1) to our paper which gives further details about eligibility for the service and a consumer profile. This Supplementary Information is referred to in the results section of the text 

All current (n=6, participation rate 100%) and two previous consumers consented to participate in the study (mean age 30.5, SD 10.9; range 18-52). Two consumers consented to the quantitative data collection (n=8 in total) but not the qualitative interviews (n=6 in total). Participants’ length of stay in the service is presented in Table 1. Consumer participants (current and previous) were typical of the consumer profile of the service (See SI 1). The diagnosis of participants included schizophrenia (n=6), schizoaffective disorder (n=1), and severe depressive disorder (n=1).

14 Line 203 – pedantic, but could you identify somewhere the schema for the quotes (clearly FG means focus group, HP health professional, CP consumer participant… but it would be good for clarity) 

 Thank you, the schema for the quotes has now been added to the third paragraph of the results section 

Participant quotes supporting identified primary themes and subthemes are denoted by: staff focus group (FG), individual interviews with staff/health professionals (HP), and consumer interviews (CP).

15 Lines 226-228 – could you elaborate on what you mean? Especially as it is unclear what ‘no actual group structure’ meant in line 231. I think it is just a bit vague and not cutting through – do you mean that due to understaffing, there was a lack of staff time to devote to developing group-based structures nor one-to-one time for consumers.? The wording has now been altered to increase clarity 

CMHRS staff emphasised that the service was ‘under-staffed’, and this resulted in them having less time to spend one on one with consumers and to devote to developing group-based structures 

16 Lines 274-276 – does this quote merit discussion also – in terms of the voluntary nature of the program – if someone is on an order, would that render them ineligible for a rehab transition to community service? Have they been left there for some other logistical convenience – no beds in acute for example? Thank you for raising this question. Some consumers are on community (medication) orders which fit within the eligibility criteria of the service. It is difficult to comment on the specific circumstances/consumers that the staff were referring to in the focus group. Therefore, we have addressed this in the discussion by suggesting that further collaboration with referrers and consumers and their families is required. We have also recommended that the referral policies and procedures be reviewed 

Therefore, further exploration of factors which may help to build communication, collaboration, and partnership between the service (management and staff), consumers, and referrers (at an organisation and individual level) is required. Revision of referral processes including policies, information, and education for consumers/families/referrers about the service (particularly in relation to its rehabilitation focus), referral documentation and procedures is needed.

17 Line 278 – I don’t understand this quote – is there context to the statement – what is the ‘this’ in ‘I was told I had to be here to get to this? Sorry?’ 

 Thank you for highlighting this we have shortened the quote to make it clearer. The quote is related to staff concerns regarding the voluntary nature of the service 

FG001.1 And sometimes they'll [consumer] say, ‘but I don't want to be here (CMHRS). I was told that I have to be here’

18 Lines 300-302 – awkward sentence – presumably an opportunity for support and the development of skills to navigate the complex systems of government support that consumers would need to engage with following discharge? Thank you, this has been changed to 

Consumers highlighted that the multidisciplinary team had assisted them to develop skills to navigate government services/systems which they would need to engage with following discharge. 

19 Line 307 – suggest that after CMHRS include ‘and recovery-oriented practice’. This has been changed as suggested 

Staff expressed concerns regarding providing transport, which they did not perceive to fit with the rehabilitative context of CMHRS and recovery-oriented practice. However, consumers indicated they found the organised transport helpful.

20 Lines 346-348 – not sure if the quote backs up the assertion – was the imbalance in the short term here? The fact that CP002 stills goes to aquafit may indicate a prioritising of supportive activities that might cause short term fatigue. Thank you we can see the wording may have been confusing and we have made some changes to increase clarity (took out reference to time frames) 

Both consumers and staff described a point-based reward system (mechanism) designed to encourage consumer motivation, thereby increasing participation in structured and unstructured rehabilitative activities. One consumer indicated this motivated them to exercise, but there were implications in terms of fatigue, which impacted on participation in an unstructured activity they enjoyed. 

21 Lines 388-390 – is there another quote to support this? The following two quotes (Lines 391-394 support the first part of the paragraph, but not the second). An additional quote has been added as suggested 

CP005: Yeah, I didn’t have much of a say, I just had to take the medication and I had to stay there for as long as it took to get the right dosage. 

22 Line 470 – I got a mean of 103(21) and 117(23) – could the authors verify their calculations and the stated figures in Table 1? (this would include the sub-domain scores too please). Thank you this has now been corrected 

There were significant positive increases in RAS-DS total scores from a mean (SD) of 102.8 (21.4) at admission to 117 (23.4) at discharge (p=.010).

23 Lines 520-522 – the results mention understaffing; however staff shortages, recruitment and retention issues are not explicitly discussed – my impression had been that perhaps the higher needs consumers were causing increased demand on the staff, thereby reducing hours available to the intended work of the program. Can these aspects be introduced and elaborated on in the results first – it is an important discussion point. Thank you, we have made a more explicit reference to the impact of staff shortages (in the results section) in response to comment 15 

CMHRS staff emphasised that the service was ‘under-staffed’, and this resulted in them having less time to spend one on one with consumers and to devote to developing group-based structures 

We have also made further reference to this in the discussion

Staff also highlighted the importance of an individually tailored rehabilitation approach, however, expressed difficulties achieving this due to staff shortages. This impacted on their ability to spend one on one time with consumers and to develop group programs and structures to address consumer goals. The service has experienced chronic issues with recruitment and retention since its inception, this has had a perceived impact on its ability to function from a rehabilitation and recovery perspective. High staff turnover does not allow for growth of knowledge and skills within the service or refinement and development of interventions and supports. Staff retention is a common issue in rural mental health services in Australia, with the following factors contributing to this – heavy workload, complexity of consumer needs, and lack of opportunity to develop a discipline specific identity and skills [17]. 

24 Lines 533, 537 – Ref 18 – could authors see newer refs by the same author that may be more appropriate - https://pubmed.ncbi.nlm.nih.gov/32295246/ and https://www.mdpi.com/1660-4601/17/8/2698

Thank you, this reference has been added as suggested

25 Lines 564-567 – I can see the tension between providing support and empowering independent use of skills (public transport on their own) – is there a reflection to be had for the rural nature of the CMHRS location and the likely relatively poor access to public transport, also for the relevance compared to what it will look like when they move on discharge (thinking specific vs general skills to navigate public transport in different places)? Thank you for this important comment. The following has been added to the discussion

Consumers come from rural towns across South Australia and the services/activities, (such as public transport, may vary greatly). These factors need to be considered when supporting consumers, and in the design and development of rehabilitation activities. 

Reviewer 2 Comments Response to Reviewer 2 Comments

1. While the authors acknowledge the likely limits of generalizability of the results, additional information about the program and participants (both staff and consumers) would help in this regard. For example, the only mention of who the program is intended for is on p. 5 (line 40), when the authors indicate that individuals referred to the program “generally have a serious mental illness and identified rehabilitative needs and goals”. These descriptors are quite vague. Additional information would be helpful. For example, the definition of “serious mental illness” can vary tremendously. For some, this refers strictly to individuals with psychotic symptoms. Others define it in regards to the level of distress and/or impairment associated with the symptoms. Thank you, to address this comment we have added the following to the manuscript:

Staff

• Table 1 which provides further detail regarding staffing (at the time of data collection and budgeted staff to show service gaps)

• Further details regarding how long staff participants had been in the service

A focus group was undertaken with six staff members (participation rate 71%) from a range of disciplines (See Table 1 for CMHRS staff profile). A majority (83%, n=5) had worked in the service for over a year (range 2.5 months to 4 years). Individual interviews were carried out with two staff members, one of whom was unable to attend the scheduled focus group (service length 4 months) and one who wanted to provide additional thoughts following the focus group session (service length 4 years). 

Consumers

Eligibility criteria and a consumer profile have been added as a Supplementary Information. This is referred to in the results section

All current (n=6, participation rate 100%) and two previous consumers consented to participate in the study (mean age 30.5, SD 10.9; range 18-52). Two consumers consented to the quantitative data collection (n=8 in total) but not the qualitative interviews (n=6 in total). Participants’ length of stay in the service is presented in Table 2. Consumer participants (current and previous) were typical of the consumer profile of the service (See SI 1). The diagnosis of participants included schizophrenia (n=6), schizoaffective disorder (n=1), and severe depressive disorder (n=1).

2. No information about the mental health history of the consumer participants was presented. While this may have been intended to help protect the confidentiality of these participants, it seems that some information could and should be provided. This would help give the reader a better understanding of the program and the identified strengths and weaknesses of it. Inspection of Table 1 suggests that the impact of the program varied substantially across participants. This seems to be the case both in relation to the RAS-DS data and levels of support required by consumers at the beginning and end of their stay. Is it possible that the impact of the program varied in relation to mental health history of the consumers? The variability in outcomes was not addressed by the researchers. While the small sample size limits the ability to reach any strong conclusions in this regard, the possibility merits consideration. Thank you this is a very interesting suggestion but was outside the scope of this study. The case note audit allowed us to collect some data i.e. diagnosis, length of stay, RAS-DS scores and details about program attendance. However, we did not have ethics approval to collect detailed information about consumers mental health history (e.g. length of illness, previous access to rehabilitation/mental health services etc)

To address your comment we have noted that this is a limitation of the study

It is acknowledged that the perceptions of all potential key stakeholders for example consumers families, referrers and case-managers (following discharge) have not been captured (as they were not part of the study inclusion criteria) and could be the focus of future research. Further exploration of consumers mental health history and impact on service engagement and RAS-DS scores may provide additional information regarding suitability for different consumer groups. Long-term outcomes for consumers (e.g. 6 to 12 months post-discharge) were not measured and this will be an important addition to ongoing service evaluations. Finally, further exploration of staff support needs and collaborative approaches (including staff, consumers and families) for designing program activities/interventions would also be beneficial.

3. In a similar vein, very little information was provided about the staff participants. Again, it is stated in the introduction (p. 5, line 41) that the team is multi-disciplinary. However, the different professional disciplines included is never described. Nor is it known if the staff who participated in the research represented the breadth of the professions and staff roles. This is pertinent as it speaks to the representativeness of the views expressed by staff members, which may strengthen or weaken some of the conclusions. Thank you this has now been addressed. See response to comment 1. The following has also been added to the results

A focus group was undertaken with six staff members (participation rate 71%) from a range of disciplines (See Table 1 for staffing profile). A majority (83%, n=5) had worked in the service for over a year (range 2.5 months to 4 years). Individual interviews were carried out with two staff members, one of whom was unable to attend the scheduled focus group (service length 4 months) and one who wanted to provide additional thoughts following the focus group session (service length 4 years).

Overall, the statistics were appropriate. Nonetheless, there were concerns about the

statistical analysis of the RAS-DS data. On p. 9, the authors indicated that they analyzed these data using the “Wisconsin paired-signed rank test”. Presumably, they meant the “Wilcoxon paired-signed rank test.” The rationale for use of this statistical procedure was not stated. However, this test is typically used when the distribution of scores deviates from a normal distribution. No information was provided about the shape or skewness of the distribution. If the distribution did, in fact, deviate from normal (which is likely given the small sample size), then the use of the mean scores (see pp. 24-25) is questionable. Generally, the median is used to summarize the data. In addition, the presentation of the results of these analyses was incomplete. While the authors indicated the associated significance levels, it would be helpful if they also stated the Z (or T) scores and an estimate of the effect size (typically, determined using rank-biserial correlations). Also, please clarify if one-tailed or two-tailed tests of significance were used. While a case could, perhaps, be made for one-tailed tests, two-tailed tests may be more appropriate since the authors are interested in what has been effective and what hasn’t. The directionality of the tests of significance should be specified and the rationale for this decision provided. Also, it was unclear why the quantitative information obtained from the audit case review was not analyzed in a similar manner. In particular, it seems that the pre- /post- level of support required could be analyzed. Thank you for your detailed comments to assist with revisions to our quantitative analysis. The Wilcoxon paired signed rank test was originally suggested by a statistician due to the small sample size. We have now re-assessed normality of data in SPSS using normality tests/plots and histograms. This was reviewed with the statistician and it was found that a majority of the data was normally distributed, therefore paired t-tests were used. There was one domain on the RAS-DS which was not normally distributed – for this domain we used Wilcoxon paired signed rank test. 

As suggested, we have also analysed pre-post levels of support. This data was also not normally distributed, and Wilcoxon paired signed rank test was used.

In the methods section (under the RAS-DS heading) the following has been added

Pre-post (on admission and discharge) scores for the RAS-DS (total raw scores and raw scores for each of the 4 domains) were analysed using paired t-tests (normally distributed data) and Wilcoxon pair signed rank tests (for data that was not normally distributed). Normality of distribution was determined using Shapiro-Wilk test of normality and histograms. Data for total raw scores and scores on 3 domains were normally distributed. Data on the ‘Doing Things I Value’ domain was not normally distributed (pre-test scores .003 on Shapiro-Wilk test).

In the methods section (under the case note audit heading) the following has been added

Case notes were examined to determine duration of stay, number and types of services received, and support required (when accessing these services). Pre-post support required (on admission and discharge) were analysed using Wilcoxon pair signed rank tests (as data was not normally distributed with a post-test score .015 on Shapiro-Wilk test)

We have added another table to support our results which includes further information regarding our pre-post analysis. The table included mean (SD) and median (IQR), Z or T scores and p values. We have also added effect size and 95% confidence intervals. The statistician was not familiar with the rank-biserial correlations, but suggested we calculate effect size by subtracting the means and dividing the result by the pooled standard deviation. 

As indicated in the response to the question concerning the quality of writing, in the main, it is written in standard English. However, the paper would benefit from careful proof-reading or use of a manuscript editing service. Sentences were often very long (e.g., 5+ lines in length) and awkwardly structured such that the writing distracted from the content of the manuscript. Revisions regarding appropriate use of the possessive and punctuation would be very helpful. The article has been proofread and edited by three members of our team and changes made throughout to address these comments

There were also some problems with the formatting of Table 1 and Figure 1. Despite several attempts to download Table 1, I was unable to download a copy that fit within the margins. Specifically, the left-hand column was partially cut-off. Please check the formatting. In a similar fashion, the resolution of Figure 1 was poor. Moreover, the size of the font was very small, making it very difficult to read. Thank you, we have now reduced the width of table 1 and re-produced Figure 1 to increase clarity

---

## [Decision Letter · Decision Letter 1]

8 Nov 2021

Demonstrating the processes and outcomes of a rural Community Mental Health Rehabilitation service: A realist evaluation

PONE-D-21-02032R1

Dear Dr. Kernot,

We’re pleased to inform you that your manuscript has been judged scientifically suitable for publication and will be formally accepted for publication once it meets all outstanding technical requirements.

Kind regards,

Giuseppe Carrà, PhD

Academic Editor

PLOS ONE

Additional Editor Comments (optional):

Reviewers' comments:

Reviewer's Responses to Questions

**Comments to the Author**

1. If the authors have adequately addressed your comments raised in a previous round of review and you feel that this manuscript is now acceptable for publication, you may indicate that here to bypass the “Comments to the Author” section, enter your conflict of interest statement in the “Confidential to Editor” section, and submit your "Accept" recommendation.

Reviewer #2: All comments have been addressed

2. Is the manuscript technically sound, and do the data support the conclusions?

Reviewer #2: Yes

3. Has the statistical analysis been performed appropriately and rigorously? 

Reviewer #2: Yes

4. Have the authors made all data underlying the findings in their manuscript fully available?

Reviewer #2: Yes

5. Is the manuscript presented in an intelligible fashion and written in standard English?

Reviewer #2: Yes

6. Review Comments to the Author

Reviewer #2: (No Response)

7. PLOS authors have the option to publish the peer review history of their article (what does this mean?). If published, this will include your full peer review and any attached files.

Reviewer #2: **Yes: **Carolyn Szostak, Ph.D.

---

## [Editor Report · Acceptance letter]

15 Nov 2021

PONE-D-21-02032R1 

Demonstrating the processes and outcomes of a rural Community Mental Health Rehabilitation service: A realist evaluation 

Dear Dr. Kernot:

I'm pleased to inform you that your manuscript has been deemed suitable for publication in PLOS ONE. Congratulations! Your manuscript is now with our production department. 

Kind regards, 

on behalf of

Dr. Giuseppe Carrà 

Academic Editor

PLOS ONE